# Optimizing CO$_2$ field flooding during sternotomy: In vitro confirmation of the Karolinska studies

**Mira Puthettu**[1,2]*, **Stijn Vandenberghe**[1,3], **Spyros Balafas**[4], **Clelia Di Serio**[3,4,5], **Geni Singjeli**[1], **Alberto Pagnamenta**[5,6,7], **Stefanos Demertzis**[1,3]

**1** Department of Cardiac Surgery, Istituto Cardiocentro Ticino, Lugano, Switzerland, **2** Laboratory of Cardiovascular Engineering, Laboratories for Translational Research EOC (LRT-EOC), Bellinzona, Switzerland, **3** Faculty of Biomedical Sciences, Università della Svizzera Italiana, Lugano, Switzerland, **4** University Centre for Statistics in the Biomedical Sciences, Vita-Salute San Raffaele University, Milano, Italy, **5** Clinical Trial Unit (CTU), Ente Ospedaliero Cantonale (EOC), Lugano, Switzerland, **6** Department of Intensive Care, Ente Ospedaliero Cantonale (EOC), Lugano, Switzerland, **7** Division of Pneumology, University of Geneva, Geneva, Switzerland

* mira.puthettu@eoc.ch

**Data Availability Statement:** All relevant data are within the paper and its Supporting information files.

## Abstract

Although CO$_2$ field-flooding was first used during cardiac surgery more than 60 years ago, its efficacy is still disputed. The invisible nature of the gas and the difficulty in determining the "safe" quantity to protect the patient are two of the main obstacles to overcome for its validation. Moreover, CO$_2$ concentration in the chest cavity is highly sensitive to procedural aspects, such suction and hand movements. Based on our review of the existing literature, we identified four major factors that influence the intra-cavity CO$_2$ concentration during open-heart surgery: type of delivery device (diffuser), delivery CO$_2$ flow rate, diffuser position around the wound cavity, and its orientation inside the cavity. In this initial study, only steady state conditions were considered to establish a basic understanding on the effect of the four above-mentioned factors. Transient factors, such as suction or hand movements, will be reported separately.

## Introduction

Air embolism during cardiac surgery has been widely reported by many studies. According to Chung et al. [1], from a few hundred to a few thousand of air bubbles enter cerebral territories during heart surgery, and these events can lead to neurological problems. In the 2015 study by Patel et al. [2], 76% of the considered patients exhibited new cerebral microbleeds, 31% new ischemic lesions and 46% experienced post-operative cognitive decline. However, in their more recent study [3], no statistically significant association was found between macro-bubbles and cerebral microbleeds. Silent cerebral ischemic lesions were also observed in transcatheter aortic valve replacement (TAVR) patients by DeCarlo et al. [4]. Bendikaite et al. confirmed that cognitive impairment is a common complication after heart surgery [5]. A meta-analysis on a total of 2632 patients done by Indja et al. [6] focused on silent brain infarcts showed that its prevalence depends on the type of surgery. Consistently to Patel et al., valve

 

**Funding:** This study was funded by Innosuisse (grant 40323.1IP-LS) and by internal funding of Istituto Cardiocentro Ticino. There was no additional external funding received for this study. The external funders had no role in study design, data collection and analysis, decision to publish, or preparation of the manuscript.

**Competing interests:** The authors have declared that no competing interests exist.

surgeries resulted in the highest prevalence compared to others: transcatheter aortic valve implantation (TAVI) 0.71, aortic valve replacement (AVR) 0.44, mixed surgery 0.39 and coronary artery bypass graft (CABG) 0.25.

For this reason, several de-airing techniques have been used for air bubble removal in the open heart and/or in the chest cavity and bubble prevention techniques were developed, such as carbon dioxide ($CO_2$) field-flooding. The principle of this technique is simple: $CO_2$ is heavier than air, so it will push air up and out of the chest cavity through the sternotomy. Moreover, $CO_2$ is much less disposed to embolize due to its higher solubility and thus forms an inert environment. $CO_2$ insufflation was firstly applied in 1958 but several doubts and concerns about its efficiency and safety remain nowadays [7]. According to the survey carried out by Orihashi et al. [8], only 36.7% of the interviewed surgeons thought that $CO_2$ insufflation was essential. Additionally, chosen $CO_2$ flow rate was observed to be a personal preference of the surgeon, where 2 L/min was the most frequently used even though 10 L/min is the flow rate suggested by literature and all diffuser companies. Due to lack of solid proof on efficacy of this technique, little knowledge on best ways to use it is available, as well as a universal definition of a "safe" threshold for intra-cavity $CO_2$ concentration. However, studies by the Karolinska Institute already showed that some factors related to the diffuser may affect the final $CO_2$ concentration in the chest, such as device design, position, depth inside the cavity, and the gas flow rate [9–11].

The goal of this study is to test all possible combinations of the four above-mentioned factors to determine optimal conditions and to have better insights on efficacy of $CO_2$ insufflation as a de-airing technique.

## Materials and methods

### SurgerySim model

A sternotomy model (model Suitcase Thorax (#1264), The Chamberlain Group, USA) was chosen to represent the chest cavity, while a complete 4-chamber rubber heart model was 3D printed (Fig 1b). The left atrium was cut open to simulate a mitral valve procedure and the model was extended with a molded silicone aorta, including the supra-aortic vessels and a portion of descending aorta. These arteries were closed off at their extremities with silicone glue to prevent $CO_2$ from entering.

Additional surgical elements were added to have a more realistic setup: an aortic cross clamp on the ascending aorta, a rib spreader, and a mannequin next to the model to represent the surgeon.

The heart model was connected to two calibrated roller pumps of a heart-lung machine (Maquet HL20, Getinge Group, Rheinfelden, Switzerland) via two suction lines: a vent (DLP Aortic Root Cannula, 9 Fr, Medtronic, Minneapolis, USA) inserted in the aortic root below the aortic clamp, and a sump (DLP Pericardial/Intracardiac Sump, 20 Fr, Medtronic, Minneapolis, USA) positioned inside the open left atrium. Flow rates were set to 300 ml/min for the vent and to 500 ml/min for the sump and these pumps were always on for all experimental runs.

The sternotomy model was placed in a Z-type Schlieren optical test bench that was used in previous studies to visualize $CO_2$ gas clouds [11,12]. An example picture taken by the system is shown in Fig 1c.

### Sensors and data acquisition

$CO_2$ flow rate was measured by a flowmeter (Mass digital flow meter, MassView, Bronkhorst High-Tech B.V., Ruurlo, NL), located downstream of the $CO_2$ tank and upstream of the diffuser.

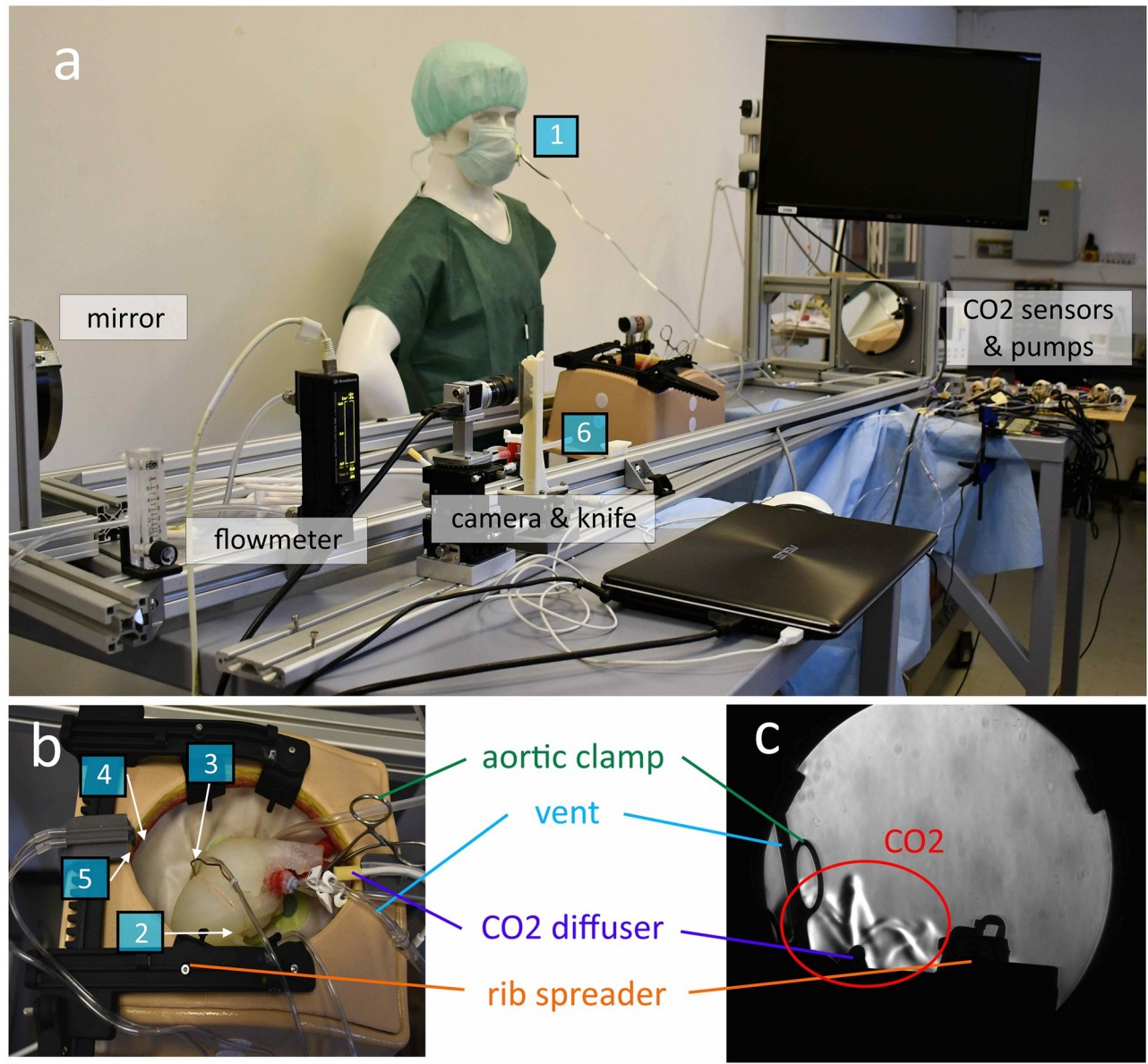

**Fig 1. SurgerySim model simulating situation in the operating room.** (a) Overview of the setup. (b) Top view of the sternotomy model. (c) View from the Schlieren's system. Gas sampling spots are marked as blue squares according to the following numbering: 1) surgeon's nose, 2) left ventricle, 3) underneath the heart, 4) at fat level, 5) at skin level and 6) at table level (overflow).

Six gas sampling lines consisting of 1.5 mm ID PVC tubes were fixed in the model to assess the local $CO_2$ concentration and they were placed in the following spots (Fig 1a and 1b): 1) at the surgeon's nose level, 2) inside the left ventricle, 3) underneath the heart, 4) 1.5 cm below the skin (fat level), 5) just above skin level, and 6) at table level next to the patient (overflow). The first two locations were fixed along all recordings, while the others were depending on the clock position of the diffuser, such that they were always on the opposite side of the diffuser before starting a new measurement.

Sampling spots outside the model (surgeon's nose and table level) were sampled continuously (0.25 L/min), while the sampling spots in or close to the heart were sampled

intermittently (0.25 L/min for 5s, then 25s off) and alternately via digital timing boards to avoid disturbance of the $CO_2$ atmosphere by excessive suction. The sampling flow in each line was generated by a membrane pump (ZR320-03PM, ZhengFuRui, Dongguan, China), which pushed the gas volume over a high-speed wide-range $CO_2$ concentration sensor (SprintIR-6S, GasSensingSolutions, Cumbernauld, UK).

All six raw concentrations and the delivery flow data were recorded in real-time (at 20 Hz and 10 Hz, respectively) with a MATLAB script (Matlab 2017b, The MathWorks Inc., Natick, MA) via USB hub with 6 ports.

### Data collection protocol and processing

The four $CO_2$ insufflation factors tested are listed in Fig 2.

Four commercially available diffuser types were included in the experiment, as well as a multiperforated drainage catheter, which is used by some surgeons as a diffuser alternative.

Three delivery flow rates were chosen according to common habits, literature, and suggestions from diffuser companies: 2, 5 and 10 L/min.

| Types of diffuser | Flow rates [L/min] | Positions | Orientations |
|---|---|---|---|
| Andocor  Temed  CarbonAid  CarbonMini  Drainage catheter | 2  5  10 | 12 (cranial)  3  6 (caudal) | Flat (0°)  Bent (90°) |
| **Total combinations** | 5x3x3x2 = 90 | | |
| **Skipped combinations** | -9 (drainage catheter cannot be positioned flat) | | |
| **Total measured combinations** | 81 | | |
| **Repetitions** | 3 | **Total runs** | 81x3 = 243 |

**Fig 2. Overview of investigated variables.** List of the four $CO_2$ insufflation factors with their corresponding levels and calculation of the total number of runs recorded.

The diffuser position around the incision was defined as a clock face, where 12 o'clock represented cranial positioning and 6 o'clock the caudal side of the patient. Only 1 lateral position (3 o'clock) was considered, the one opposing the mannequin side.

Two insertion orientations were tested based on observations of various surgeons: a 0˚ angle or flat orientation where the diffuser was fixed parallel to the skin of the patient with the tip hanging (~horizontally) over the cavity, and a 90˚ angle or bent orientation where the diffuser was inside the cavity with the tip aligned vertically and with a depth of ~2 cm.

All combinations of these factors were tested, except for 9 of them. Due to its geometry, a flat orientation could not be tested with the drainage catheter, where $CO_2$ is released mainly from its lateral holes, so it was always inserted deep in the cavity to have all holes below skin level.

For each measured combination, $CO_2$ concentrations were recorded during five minutes (run), where the start of $CO_2$ delivery was synchronized with the recording start, resulting graphically in a hyperbolic curve, beginning at 0% of $CO_2$ and then slowly increasing until reaching a stable value (plateau).

Three repetitions of all measured combinations were performed, where one repetition consisted of 81 runs of unique factor combinations in a randomized order.

Between one run and the other, after stoppage of insufflation and $CO_2$ concentrations recording, remaining $CO_2$ was removed from the model and all possible collecting spots by a vacuum pump. Purging was terminated only when all $CO_2$ sensors registered a level of $CO_2 <$ 1%. At this point delivery device and non-fixed sampling spots were positioned according to the next combination to be measured.

After recording, all signals were filtered by a MATLAB script that applies a Yule-Walker low pass IIR filter of order 50 (cut-off frequency of 50 mHz). After filtering, the mean $CO_2$ concentration of the plateau (last three minutes of the run) was calculated for each combination and all sampling spots.

## Statistical analysis

Goal of the analysis was to select important explanatory factors that affect the $CO_2$ concentration levels at i) underneath the heart, and ii) the left ventricle. We modelled the $CO_2$ concentration levels from each area separately, using a hybrid estimation method. In particular, the analysis routine consisted of two steps in which first we fit a penalized linear mixed model (pLMM) for performing variable selection, and then a linear mixed model (LMM) was fitted to the data including only the selected variables.

The LMM [13,14] is common choice for modelling clustered data such as repeated measures. For selecting the important explanatory variables for the $CO_2$ concentration levels we employed the simultaneously penalized linear mixed model (SPLMM) [15, chap. 3]. The latter uses the least absolute shrinkage and selection operator (LASSO) [16] or ℓ1−regularization to shrink the absolute size of the model parameters towards zero for performing simultaneous selection of fixed and random effects. In the first analysis step, as fixed effects we considered p = 11 dummy covariates corresponding to i) different types of diffusers, ii) different types of $CO_2$ flow rate delivery method, iii) diffuser position around the wound cavity, and iv) orientation of the diffuser inside the cavity.

The random effects part of the model included a single random intercept. The optimal model was selected by minimizing the Bayesian information criterion (BIC) [17]. After the variable selection step, the model from the first step was refitted using a traditional LMM including the selected parameters. The statistical analysis in this paper has been performed in the R software version 4.2.1 [18] using statistical packages therein. Particularly, for performing

simultaneous selection of fixed and random effects the splmm package [19] was used. For getting uncertainty estimates of the selected parameters, the final regularized model was refitted using the lme4 package [20].

### Ethics statements

This article does not contain any studies with human or animal subjects. Ethical approval is not applicable for this article.

## Results

A first phase was dedicated to graphically exploring the data.

No trend was identified for sampling spots at skin level (range: 0.18–78.8%) and overflow (range: 0.06–22%). Concentrations at the level of the surgeon's nose ranged from 0.04% (CO$_2$ percentage in air) to 0.18% (drainage catheter at 10 L/min in position 3), which is close to the CO$_2$ sensors accuracy range. For this reason, a focus was set on the remaining intra-thoracic sampling spots (at fat-level, left ventricle and underneath the heart), while the others were neglected during the analysis.

Fig 3 shows how maximum achievable CO$_2$ concentration (plateau) varies as function of flow rate for each type of diffuser. This is shown for both orientations (flat/bent) and for the three considered sampling spots. Data points correspond to marginal means obtained over the three positions around the wound cavity. An arbitrary threshold of 90% CO$_2$ was defined as a "safe" level for a better comparison between configurations (split shown by the grey area).

These plots show that when the diffuser was positioned flat, mean concentrations varied from 49% (Temed, 2 L/min, at fat level) to 87% (Andocor and CarbonAid, 10 L/min, in the left ventricle). A concentration above 90% was never reached (on the left, curves are always in the grey area). This is also true for the multiperforated drainage catheter which, in contrast with all other trends, reached its maximum CO$_2$ concentration with the lowest flow rate of 2 L/min (77% in the left ventricle), while dropping to 42% at the maximum tested flow rate of 10 l/min. Commercial diffusers always exhibit their lowest concentration at the lowest flow rate.

While using a commercial diffuser at bent angle, the CO$_2$ concentration obtained was always equal to or above 90% with no big effect of flow rate. The exception was at fat level, so near the top of the chest cavity, at 2 L/min flow, where concentrations dropped to the 60–80% range. Additionally, differences between commercial diffusers were minimized in this orientation (standard deviation range: 1.5–2.9%, expect for 6.9% at 2 L/min at fat level).

The same exploration was done for the factor of position around wound cavity. Fig 4 shows how CO$_2$ concentration varies as function of position for each type of diffuser. Data points correspond to marginal means obtained over the three flow rates.

Once again when commercial diffusers were positioned at flat angle and for the drainage catheter, CO$_2$ concentrations were always below 90%. The highest concentrations were reached at position 12 (cranial, CO$_2$ range in the left ventricle: 68–87%) whereas the lowest ones were obtained at position 6 (caudal, CO$_2$ range in the left ventricle: 50–74%). For commercial diffusers at bent angle, the effect of position was not evident in the left ventricle and underneath the heart. In this case, mean concentrations ranged from 78% (at fat level, position 12) to 98% (in the left ventricle, position 12), for all flow rates combined.

Also in this case, differences between commercial diffusers were minimized at bent angle (standard deviation range: 0.8–3.3%).

Fig 5b shows Schlieren's pictures for all commercial diffusers at flat angle (all flow rates). These pictures depict only what happened above the cavity and on lateral sides (view from

## Effect of flow rate on CO2 concentration

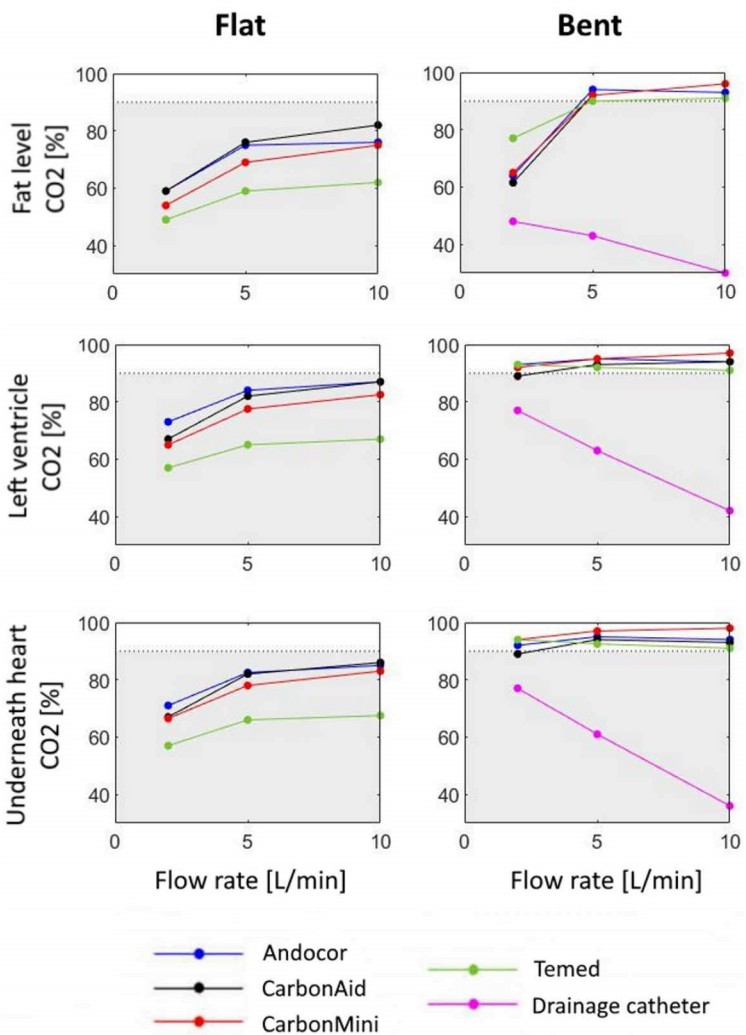

**Fig 3. Marginal means of $CO_2$ concentration for all types of diffusers as function of flow rate.** Trends are shown for both orientations and the three considered sampling spots. Concentrations below 90% are in the grey area.

caudal to cranial side). A portion of the $CO_2$ gas could be observed to flow outside the thorax cavity. While Temed and CarbonMini clearly exhibited jets of $CO_2$ gas spraying in all directions, Andocor and CarbonAid appeared to have a more diffuse delivery of the gas for all flow rates, allowing the gas to fall in the cavity without creating turbulence.

Fig 6 shows Schlieren's pictures for all commercial diffusers and the drainage catheter at bent angle (all flow rates). In contrast to the previous figure, when any of the commercial diffusers was bent inside the cavity, no more gas clouds were spawn in the atmosphere above the chest cavity and thus all the $CO_2$ gas was delivered inside. The overflow was much more contained; in this case, $CO_2$ was visible only at surface of the model. While for the drainage catheter an important turbulent cloud is observed above the wound cavity.

Schlieren's pictures confirmed also that differences between commercial diffusers are minimized when they are bent inside the chest cavity.

## Effect of position on CO2 concentration

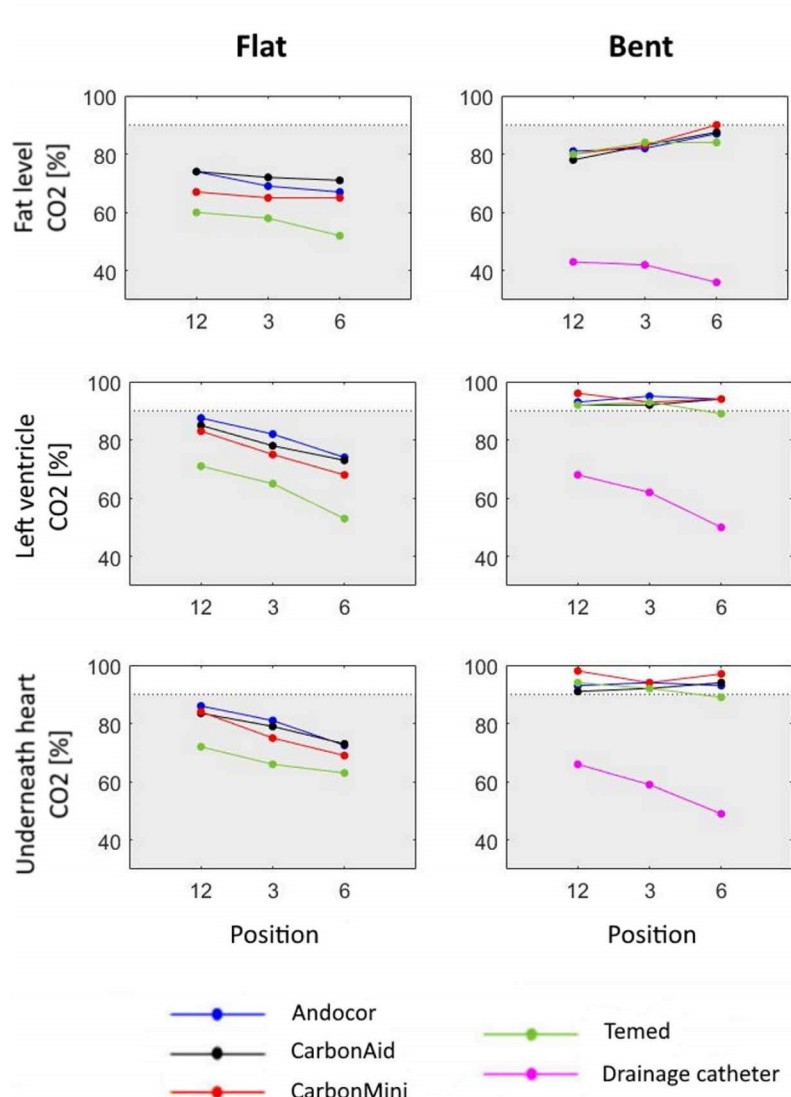

**Fig 4. Marginal means of CO$_2$ concentration for all types of diffusers as function of position.** Trends are shown for both orientations and the three considered sampling spots. Concentrations below 90% are in the grey area.

Fig 7 shows an overview of all combinations. The arbitrary threshold of 90% is used again for a better comparison between mean values of the three repeated measures.

In a second phase, these preliminary observations were tested for statistical significance.

Table 1 shows the main conclusions from the SPLMM. Focus was set on the left ventricle and underneath the heart, which are the priority areas to be de-aired during cardiac surgery. The complete table with statistical inference results is provided in the Supporting information section.

Statistical results confirmed that bending the diffuser inside the wound cavity had a strong positive effect on CO$_2$ concentration levels (leading to an increase) both inside the left ventricle and underneath the heart. The estimated fixed effect coefficients (β) for bending the diffuser were 18.83 and 19.37 (p < 0.0001; see Supporting information), respectively.

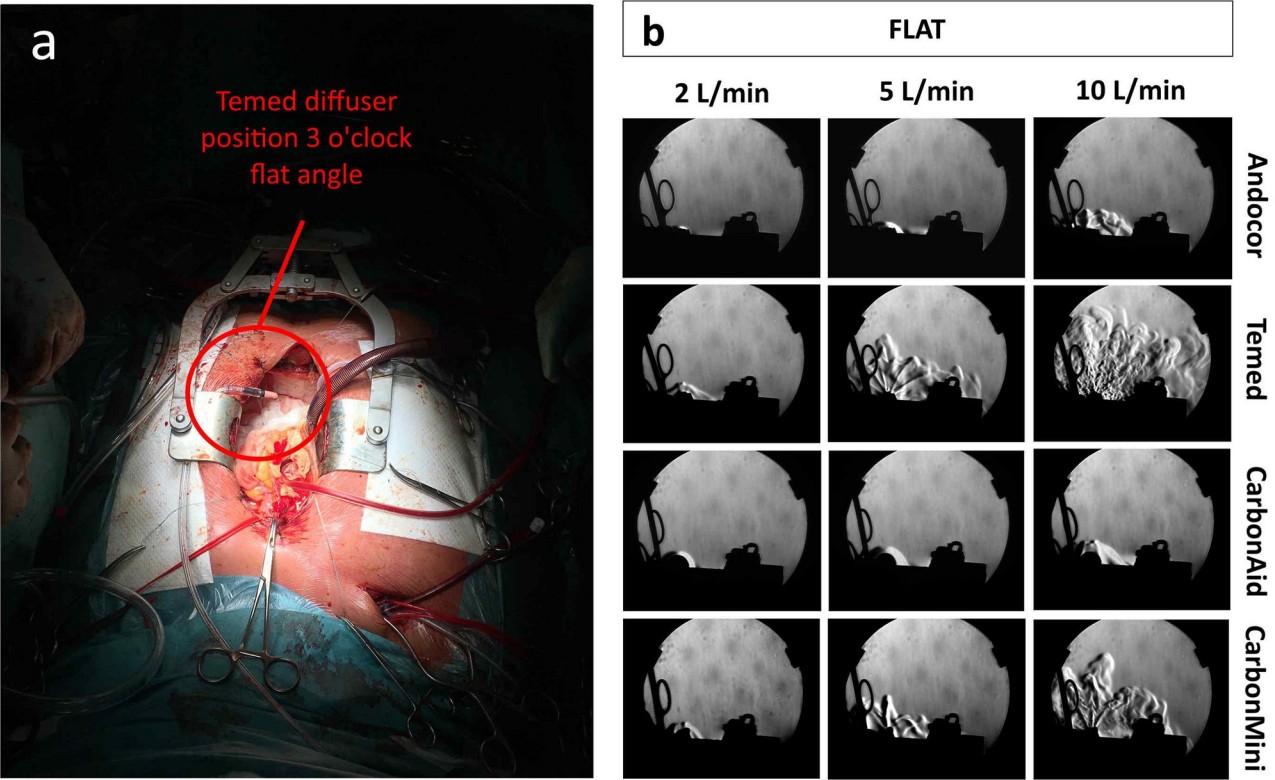

**Fig 5. Schlieren's pictures for flat orientation.** (a) Picture of an example case in the operating room where the diffuser (Temed) was positioned at flat angle and at position 3 o'clock. (b) Comparison between commercial diffusers at flat angle for all flow rates in the in vitro sternotomy model. Pictures taken with the Schlieren's system (view from caudal to cranial).

A negative effect on $CO_2$ levels was observed for the drainage catheter ($\beta$ = -15.80; $p < 0.0001$) and Temed ($\beta$ = -7.69; $p < 0.01$). However, as mentioned previously, based on the explorative plots, difference between Temed and the other commercial diffusers was minimal at bent angle.

Another determinant was position 6 (caudal) which reduced $CO_2$ levels inside the left ventricle ($\beta$ = -4.75; $p < 0.05$) and underneath the heart ($\beta$ = -4.98; $p < 0.05$).

## Discussion

To this current day, efficacy of $CO_2$ field-flooding remains to be proven. However, this is not an easy task due to the lack of standardized application method, the invisible nature of the gas and the complex dynamics in the operating room during cardiac surgery.

Due to the high sensitivity of $CO_2$ to external factors and the variability in the way $CO_2$ field-flooding is performed, there are a lot of discrepancies between previous studies. Because of all these reasons, it is also difficult to determine a "safe" threshold above which $CO_2$ concentration in the heart or in the chest cavity guarantees complete prevention of air emboli. With this limited current knowledge, the best approach for the time being is to get the highest $CO_2$ concentration possible by optimizing the way $CO_2$ insufflation is performed.

A research group at the Karolinska Institute previously examined the effects of different factors such as diffuser type, flow rate, and position on $CO_2$ field-flooding. Our results are very

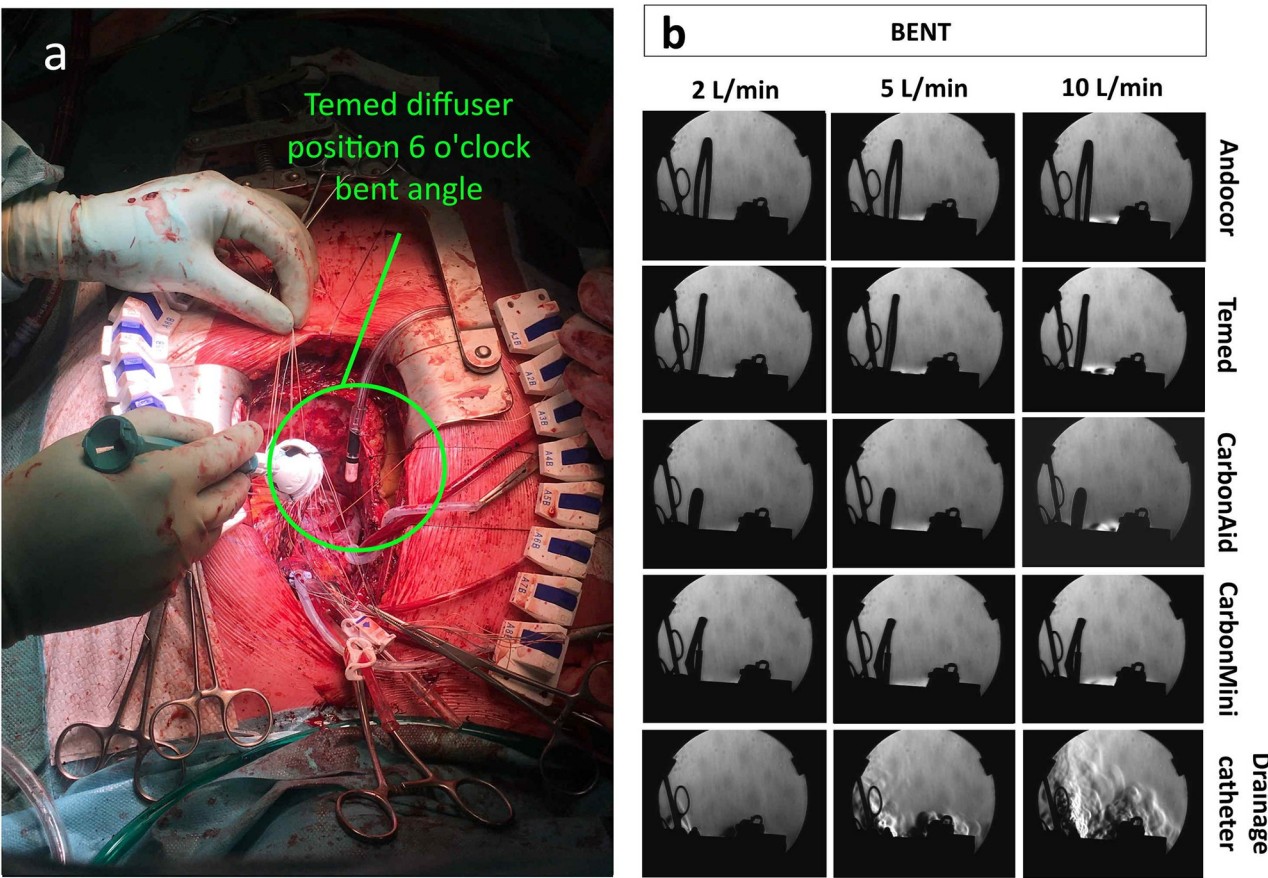

**Fig 6. Schlieren's pictures for bent orientation.** (a) Picture of an example case in the operating room where the diffuser (Temed) was positioned at bent angle and at position 6 o'clock (caudal). (b) Comparison between commercial diffusers and the drainage catheter at bent angle and for all flow rates in the in vitro sternotomy model. Pictures taken with the Schlieren's system (view from caudal to cranial).

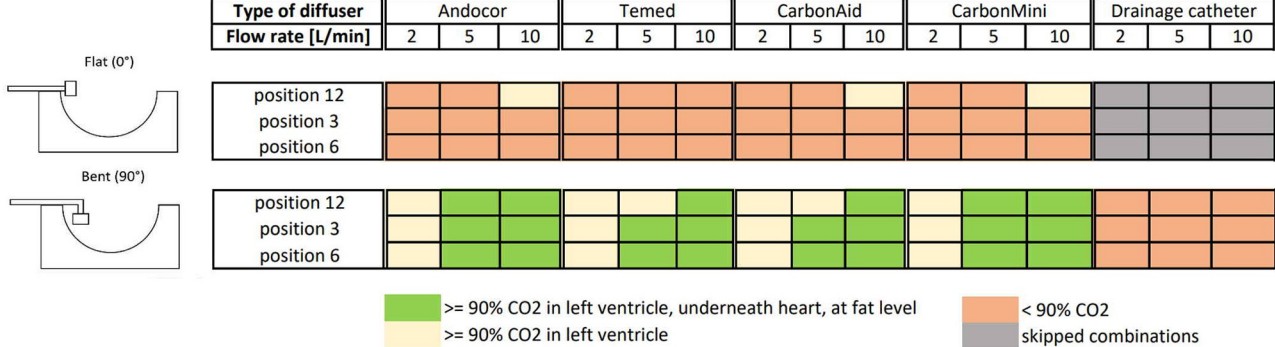

**Fig 7. Summary of the explorative phase.** Combinations reaching at least 90% of CO$_2$ in all three considered sampling spots (left ventricle, underneath the heart and at fat level) are shown in green, those reaching 90% only in the left ventricle are shown in yellow, while the others are colored in orange. Skipped combinations are colored in dark gray.

**Table 1. Selected variables with corresponding effect on CO₂ concentration.**

| Left ventricle / underneath heart | |
|---|---|
| Bent | ++ |
| Drainage catheter | - - |
| Temed | - |
| Position 6 | - |

-, negative (decrease in CO₂ concentration; -10 < = β < 0); - -, strongly negative (β < -10); +, positive (increase in CO₂ concentration; 0 < β < = 10); ++, strongly positive (β > 10); where β is the estimated fixed effect coefficient.

consistent with their various published reports, albeit with a different model and a different measurement setup.

In their study published by Persson et al. in 2003 [9], a gas diffuser was compared to conventional PVC tubes in controlled static conditions at four different flow rates (2.5, 5, 7.5 and 10 L/min). In all cases, the diffuser was in fixed position 4 cm inside the chest cavity. They concluded that the gas diffuser performed significantly better than PVC tubes. In our study, we observed that commercial gas diffusers also perform better than a multiperforated drainage catheter. According to our observations, we discourage the use of this last one, since it resulted in a strongly negative effect on CO₂ levels and it generated a turbulent cloud of gases above the chest cavity, which indicates a mixing of air and CO₂. This turbulence likely entrains air into the chest cavity and thereby reduces the efficacy of field-flooding, which is corroborated by the decrease in our concentration data.

In the same study by Persson et al. [9], they observed that a delivery flow rate of at least 5 L/min is necessary to counterbalance diffusion, but they hypothesized that higher flows may be needed during real cardiac surgery to compensate for turbulence generated by hand movements, convective air currents from ventilation and use of suction. Optimal conditions in their study were reached with a flow rate of 7.5 L/min. In our study, effect of flow rate was dependent on the tested orientation. In bent orientation, flow rate and position of diffuser seemed not to influence the CO₂ concentration, mainly because it was already near its maximum level of 100%. While in flat orientation, an increase in flow rate could boost the CO₂ concentration for commercial diffusers, while with the drainage catheter the inverse was true.

In a 2004 study from the Karolinska group [10], four flow rates (2.5, 5, 7.5 and 10 L/min) were tested at different delivery depths: at the top level of the wound cavity and inside the wound cavity (5 cm below). All devices were more efficient when positioned inside the cavity. This was also confirmed by our tests, where bending the gas diffuser resulted in a strongly positive effect on CO₂ levels. For commercial diffusers, by simply bending the tip of the device at least 2 cm deep inside the chest cavity, an increase in CO₂ concentrations was reported for all cases with an average jump of 18.8% in the left ventricle, reaching a mean concentration of 93.4%. While in flat orientation, the average CO₂ concentration in the left ventricle was only 74.6%.

Interestingly, the Karolinska group did not investigate the effect of positioning around the wound cavity; in fact, in most of their studies, the diffuser was installed caudally (above the diaphragm), while we can conclude that a cranially placed diffuser performs better. From our results, statistical results identified caudal position (position 6) as the least performant. No difference was found between positions 12 and 3 o'clock. However, we are aware that the diffuser should be positioned in a way that allows the surgeon to operate comfortably and we have observed all three positions being used during various surgeries.

Our SurgerySim model has a couple of major advantages. It was inspired by previous models used in the literature and a high degree of anatomical realism was added. Compared to

other simulated environments, our model included two suction lines from a real heart-lung machine that are always present during on-pump cardiac surgery and that were neglected in previous studies. Some previous studies tested CO$_2$ insufflation in an empty chest cavity, while we included a detailed heart model to faithfully reproduce the filling by CO$_2$ gas via an atrial incision. We also use the Schlieren system to have actual visual proofs on how CO$_2$ behaves in such conditions. In fact, Schlieren's recordings allowed us to spot differences in the CO$_2$ jets between types of diffusers. Temed and CarbonMini performed slightly worse than the other commercial ones in flat orientations (significant only for Temed), but this is probably explained by their strong jets all around the tip. Because of this design, CO$_2$ was propelled far away with higher flow rates, even outside the chest cavity, while for Andocor and CarbonAid an almost laminar flow was observed also at 10 L/min.

However, our in-vitro study has some clear limitations. First, all measurements were performed in the same "patient", which means that factors such as anatomy, size of the heart and cavity, type of surgery (with different incision location) were neglected. Moreover, the modeled environment was in dry conditions where there was no blood, no tissues, and no organs, so absorption of gases was not considered. Additionally, for this first exploratory study, CO$_2$ concentrations were measured in static conditions, which is not representative for a real cardiac surgery. We have previously reported in a simplified model on the effect of hand movements of the surgeon, which generate slight turbulence, and on the effect of suction, which has a major impact on the achievable CO$_2$ concentration [11]. Such disturbing factors will also be investigated in this newer, more accurate model now that we have established the main determinants of steady state performance. Effect of in-room ventilation was also tested in the lab room and in an actual OR during the preliminary phase, but no significant impact was observed, so the current study was performed with disabled ventilation. Another limitation was the arbitrary choice of a threshold (90% of CO$_2$) to identify "safe" combinations, due to lack of a standard, consensus, or even basic knowledge around this topic. As mentioned previously, the safest standard of care until proven differently is to achieve the highest CO$_2$ concentration possible in the thorax. For this reason, combinations that reached at least 90% are the best candidates for optimization of CO$_2$ field flooding.

Some questions may arise on the potential risks for the patient of increasing concentration of CO$_2$ in the blood. Possible systemic effects include higher levels of hypercapnia, altered cerebral hemodynamics and red blood cell damage [21]. However, severe hypercapnic acidosis can be prevented if compensatory action by the perfusionist is taken early and in an effective way. Concerning safety of the surgical staff, in our study, the maximum concentration registered at the surgeon's nose height was 0.18%. We hypothesize that changes in concentrations at this level are negligible, but assessment with more precise sensors is necessary for confirmation.

Based on our observations, we strongly suggest the use of commercial diffusers and avoid improvised devices such as PVC tubes or a drainage catheter for CO$_2$ insufflation. We also recommend bending the tip of the commercial diffuser at least 2 cm deep inside the chest cavity and to set the delivery CO$_2$ flow to at least 5 L/min (even higher to counterbalance effect of suction and hand movements). If bending is not possible due to practical reasons during surgical procedure, we suggest following the Instructions For Use of the chosen diffuser's type, since optimal flow rate and positioning may vary between different brands.

## Supporting information

**S1 File. R code for statistical analysis.** This is the R code used to run the statistical analysis on the collected data.
(TXT)

**S2 File. Input file for statistical analysis.** This file contains CO$_2$ concentration mean values for all the sampling spots obtained during plateau phase for each recording. This table was used as input for the statistical analysis in R.
(CSV)

**S1 Table. Selected fixed effects and corresponding parameters (β) for each sampling spot with their associated uncertainty estimates and p-values.** β: coefficients of selected parameters, SE: standard error, df: degree of freedom, t: t value, p: p-value, SD(ID): standard deviation of random effects variances.
(DOCX)

## Acknowledgments

We would like to thank perfusionist Gianluca Agus for the assistance with the heart-lung machine. We thank Franco Dossena for the photography of the setup and Linda Bernasconi for the pictures taken in the operating room. We also thank Dr. Michele Gallo for the preliminary testing of the model and techniques in an OR setting.

## Author Contributions

**Conceptualization:** Mira Puthettu, Stijn Vandenberghe, Geni Singjeli, Stefanos Demertzis.

**Data curation:** Mira Puthettu, Spyros Balafas, Clelia Di Serio, Alberto Pagnamenta.

**Formal analysis:** Mira Puthettu, Spyros Balafas, Clelia Di Serio, Alberto Pagnamenta.

**Funding acquisition:** Stijn Vandenberghe, Stefanos Demertzis.

**Investigation:** Mira Puthettu, Stijn Vandenberghe, Spyros Balafas, Geni Singjeli.

**Methodology:** Mira Puthettu, Stijn Vandenberghe, Spyros Balafas, Clelia Di Serio, Geni Singjeli, Alberto Pagnamenta.

**Project administration:** Stijn Vandenberghe.

**Resources:** Stijn Vandenberghe, Stefanos Demertzis.

**Software:** Spyros Balafas, Geni Singjeli.

**Supervision:** Stijn Vandenberghe, Clelia Di Serio, Stefanos Demertzis.

**Validation:** Mira Puthettu, Stijn Vandenberghe, Geni Singjeli.

**Visualization:** Mira Puthettu, Stijn Vandenberghe, Geni Singjeli.

**Writing – original draft:** Mira Puthettu.

**Writing – review & editing:** Mira Puthettu, Stijn Vandenberghe, Spyros Balafas, Clelia Di Serio, Geni Singjeli, Alberto Pagnamenta, Stefanos Demertzis.

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
