## [Decision Letter · Decision Letter 0]

21 Mar 2023

PONE-D-22-34227Optimizing CO2 field flooding during sternotomy: in vitro confirmation of the Karolinska studiesPLOS ONE

Dear Dr. Puthettu,

Thank you for submitting your manuscript to PLOS ONE. After careful consideration, we feel that it has merit but does not fully meet PLOS ONE’s publication criteria as it currently stands. Therefore, we invite you to submit a revised version of the manuscript that addresses the points raised during the review process.

We look forward to receiving your revised manuscript.

Kind regards,

Mohamad Khair Abou Chaar

Academic Editor

PLOS ONE

Journal Requirements:

"This study was funded in part by Innosuisse (grant 40323.1IP-LS)."

"This study was funded in part by Innosuisse (grant 40323.1IP-LS)."

6. We note that Figures 1, 4, 5 and Table 1 in your submission contain copyrighted images. All PLOS content is published under the Creative Commons Attribution License (CC BY 4.0), which means that the manuscript, images, and Supporting Information files will be freely available online, and any third party is permitted to access, download, copy, distribute, and use these materials in any way, even commercially, with proper attribution. For more information, see our copyright guidelines: http://journals.plos.org/plosone/s/licenses-and-copyright.

a. You may seek permission from the original copyright holder of Figures 1, 4, 5 and Table 1 to publish the content specifically under the CC BY 4.0 license. 

Additional Editor Comments:

Dear authors,

I hope this find you well. I want to start by praising your hard work. Based on the feedback from the reviewers, we kindly ask you to resubmit the manuscript after making the required changes/modifications.

Thank you,

Moe

Reviewers' comments:

Reviewer's Responses to Questions

**Comments to the Author**

1. Is the manuscript technically sound, and do the data support the conclusions?

Reviewer #1: Yes

Reviewer #2: Yes

Reviewer #3: Yes

2. Has the statistical analysis been performed appropriately and rigorously? 

Reviewer #1: Yes

Reviewer #2: Yes

Reviewer #3: I Don't Know

3. Have the authors made all data underlying the findings in their manuscript fully available?

Reviewer #1: Yes

Reviewer #2: Yes

Reviewer #3: Yes

4. Is the manuscript presented in an intelligible fashion and written in standard English?

Reviewer #1: Yes

Reviewer #2: Yes

Reviewer #3: Yes

5. Review Comments to the Author

Reviewer #1: The authors have done an in-vitro study. The same authors have previously published in-vitro data from a less sophisticated model. The clinical question that drive such studies, is whether carbon dioxide flooding during sternotomy can prevent or reduce the incidence of air embolism to the brain (and heart) during open cardiac surgery, and that this could improve clinical outcome for patients. As pointed out by the authors, even though CO2-flooding has been used for decades, doubts and concerns about its efficiency remain.

The present study tries to explore some principles of fundamental character. The model employed has definitive limitations as described by the authors, but none the less, some factors that may be prerequisite in order to obtain adequate concentrations of CO2 in and around the heart during surgery are described. This should be of interest for those concerned with the technical aspects of this method, and therefore merit publication. Whether obtaining such concentrations of CO2 is of importance in clinical medicine is a much harder question to undertake, and not the scope of this study.

Reviewer #2: This is a basic study measuring CO2 levels during cardiac surgery using a variety of catheters/diffusers and a variety of CO2 flow rates. The presumption with this study is that cerebral blood flow can be interrupted by air bubbles that escape the normal deairing process and due to the insolubility of nitrogen which makes up nearly 80% of room air and can result in prolonged obstruction to the flow of blood and that flooding the surgical field with CO2 can substantially shorten the period of blood flow interruption due to its high solubility. Overall, the benefits of CO2 flooding are controversial at best with most cardiac centers not routinely utilizing the technique. Yet the benefit may be from inadequate knowledge of the effects of various factors so this manuscript may lead to increase clinical use of this technique. While the experiments and analysis are described in adequate detail, some issues/comments arise when reading the manuscript:

1. The introduction is too long and should be shortened and simplified for this relatively limited dataset and report.

2. To increase the interest, I would suggest including data regarding the effect of the use of pump suckers on the CO2 levels as this manuscript has fairly limited data but serves as a basis for additional testing (as you indicated) but some additional variables would make this basic report much more interesting.

3. There going to be a large number of variables ultimately affecting the CO2 levels measured in this report, such as pump suckers, the surgeon and assistants and others. For instance, what was the ventilation system of the room used? OR requirements include air exchange rates which generally exceed those of normal rooms and the presence of OR lights overhead, etc. can affect the local air flow; did you perform these tests in a functional OR room or somewhere else and how would you propose to evaluate the room ventilation effect and light positions on CO2 levels?

4. Do you have a plan to monitor CO2 levels during actual surgery? How would you propose to do that? Is it practical and if so, would low CO2 levels inappropriately pressure the surgeon to change techniques?

5. What about using animal models, such as porcine models to better understand how having real tissue in an accurate anatomic configuration affects CO2 levels?

6. Overall the English grammar is acceptable but there are clear opportunities to improve this aspect and should be carefully reviewed by someone fluent in English.

Reviewer #3: Thank you for this clear article that provides interresting informations about CO2 field flooding during sternotomy. For me, one major questions remains unclear, is there an upper level of CO2 that is dangerous? Is it describe? If yes, what are the sides effects? This reflection should be a part of the discussion.

6. PLOS authors have the option to publish the peer review history of their article (what does this mean?). If published, this will include your full peer review and any attached files.

Reviewer #1: No

Reviewer #2: No

Reviewer #3: No

---

## [Author Response · Author response to Decision Letter 0]

4 May 2023

We really appreciate the efforts of the academic editor and the reviewers in improving our manuscript and the positive feedback provided during the review process. We modified the script the best we could following the reviewers’ comments and suggestions, and we offer the following point-by-point replies:

Reviewer #1

The reviewer’s comments captured very well our intention to deliver insights, while knowing that we cannot provide perfect answers for the clinical situation. We are grateful for his wise assessment that our study should only be interpreted and judged within this intended scope.

Reviewer #2

We are grateful for the reviewer’s statement that our study may help alleviate inadequate knowledge about the effects of various factors on CO2, which could lead to more consistent or equipoised clinical studies in the future.

1. The introduction is too long and should be shortened and simplified for this relatively limited dataset and report.

We removed some sentences from the introduction while keeping the key message. 

2. To increase the interest, I would suggest including data regarding the effect of the use of pump suckers on the CO2 levels as this manuscript has fairly limited data but serves as a basis for additional testing (as you indicated) but some additional variables would make this basic report much more interesting

Investigation of the effect of pump suckers is interesting as indicated by the reviewer and essential according to us. We already did it in a follow-up study, which will be presented in an upcoming article. We decided to split into two articles because the type of analysis performed is completely different and cannot be merged into one statistical analysis. For the study here under review, we assessed steady conditions to create insights in what can ideally be achieved and the whole complicated multi-factorial statistical analysis was performed on the mean CO2 values obtained during the plateau phase (equilibrium). While for the effect of pump suckers, we investigated some dynamic parameters to capture the transient effects such as rising time to reach the maximum CO2 plateau, drop in CO2 concentration due to suction, and drop and recovery percentages and times when suction is turned off and then on again. We hope that the upcoming manuscript will satisfy the curiosity of the reviewer as it complements this first article. 

3. There is going to be a large number of variables ultimately affecting the CO2 levels measured in this report, such as pump suckers, the surgeon and assistants and others. For instance, what was the ventilation system of the room used? OR requirements include air exchange rates which generally exceed those of normal rooms and the presence of OR lights overhead, etc. can affect the local air flow; did you perform these tests in a functional OR room or somewhere else and how would you propose to evaluate the room ventilation effect and light positions on CO2 levels?

We previously tested the effect of ventilation both in an actual OR (with also the surgeon wearing sensors) by moving the OR table relative to the downflow and in our lab by adjusting the ceiling fans. Preliminary observations with CO2 sensors and Schlieren visualization showed that the impact is negligible, so we performed the final study in a laboratory with disabled ventilation. We did not test the effect of heating caused by lamps, but we assume that is also negligible compared to the factors that were under investigation. Effect of surgeon and assistant was also tested in a previously published study and in our OR simulation, and we excluded it from this one due to the focus on steady conditions. 

4. Do you have a plan to monitor CO2 levels during actual surgery? How would you propose to do that? Is it practical and if so, would low CO2 levels inappropriately pressure the surgeon to change techniques?

As the reviewer guessed, we do indeed plan to monitor CO2 during actual surgery, and we are developing a prototype device to do so. Reporting on this device is premature, as it is still work in progress and therefore we considered this issue as beyond the scope of the here reported study. 

5. What about using animal models, such as porcine models to better understand how having real tissue in an accurate anatomic configuration affects CO2 levels?

We considered the option of using animal models as a next step now that we have reached insights and we can better filter the experimental settings that would yield interesting results. Since the here presented extensive set of conditions took several days of model measurements, this would be impossible to achieve in a single animal, thus weakening the statistical power. An animal model would also eliminate the option of obtaining the Schlieren insights. This would be a completely different study and beyond our initial scope, hence we choose to not discuss this in our manuscript. 

6. Overall the English grammar is acceptable but there are clear opportunities to improve this aspect and should be carefully reviewed by someone fluent in English.

We thank the reviewer for this suggestion, and we had the manuscript reviewed for language. 

Reviewer #3

1. For me, one major questions remains unclear, is there an upper level of CO2 that is dangerous? Is it describe? If yes, what are the sides effects? This reflection should be a part of the discussion.

As suggested by reviewer #3 we added a paragraph about side effects of CO2 insufflation in the Discussion section. The maximum CO2 concentration that can be reached in the atmosphere in and around the chest is 100%. The potential danger comes from its absorption into the blood, which can and should be easily compensated by an experienced perfusionist by increasing the fresh gas flow through the oxygenator. Monitoring of blood gases around the start of CO2 field flooding is part of our routine clinical practice and we have had no issues with excessive pCO2 thanks to this simple compensation method.

We also noted that reviewer #3 responded with “I don’t know” to the assessment of the appropriateness of our statistical analysis. We would like to point out that this seemingly simple laboratory parameter study has baffled highly experienced clinical statisticians due to the combination of factors, the repetitions, and the fact that our “population” consists of one patient model which reduces variability but is contrary to how clinical trials are set up. We eventually teamed up with 3 professional experts and had several discussions on approaches and models. We are convinced that the eventual presentation reflects the best way to analyze these data, but there is not unique “correct” way for this, just different acceptable approaches. 

The funding statement was corrected as requested and the cover letter was amended accordingly. Copyrighted images in Table 1 were replaced with our own. While Figures 1, 4 and 5 were kept the same because they contain only original pictures taken by our group. However, tables containing images are now labeled as figures and the numbering of figures and tables was adapted. 

Additionally, we made the necessary changes to comply with the PLOS ONE style templates. 

We would like also to request a change in the authors’ list. We would like to add a new co-author, Alberto Pagnamenta, who was previously mentioned in the Acknowledgments. After careful consideration, we conclude that Alberto Pagnamenta satisfies the requirements to be a co-author of this manuscript. All co-authors agree with this change. 

We respectfully hope that it might be possible for you to take the revised version of our manuscript into consideration. We appreciate the comments and suggestions of the reviewers and thank them for helping us to improve our manuscript.

---

## [Decision Letter · Decision Letter 1]

27 Sep 2023

Optimizing CO2 field flooding during sternotomy: in vitro confirmation of the Karolinska studies

PONE-D-22-34227R1

Dear Dr. Puthettu,

We’re pleased to inform you that your manuscript has been judged scientifically suitable for publication and will be formally accepted for publication once it meets all outstanding technical requirements.

Kind regards,

Mohamad Khair Abou Chaar

Academic Editor

PLOS ONE

Additional Editor Comments (optional):

It brings me great pleasure to let you know that we acquired the needed number of reviewers to proceed with the decision making process and it is my honor to let you know that your manuscript is accepted for publications. Your work was very interesting to read about and I believe that your manuscript will have an impact on clinical practice.

From all of us at PLOS ONE, we congratulate you and your team for this tremendous work.

Mohamad K. Abou Chaar, MD

Reviewers' comments:

Reviewer's Responses to Questions

**Comments to the Author**

1. If the authors have adequately addressed your comments raised in a previous round of review and you feel that this manuscript is now acceptable for publication, you may indicate that here to bypass the “Comments to the Author” section, enter your conflict of interest statement in the “Confidential to Editor” section, and submit your "Accept" recommendation.

Reviewer #1: (No Response)

Reviewer #3: All comments have been addressed

2. Is the manuscript technically sound, and do the data support the conclusions?

Reviewer #1: Yes

Reviewer #3: Yes

3. Has the statistical analysis been performed appropriately and rigorously? 

Reviewer #1: Yes

Reviewer #3: I Don't Know

4. Have the authors made all data underlying the findings in their manuscript fully available?

Reviewer #1: Yes

Reviewer #3: Yes

5. Is the manuscript presented in an intelligible fashion and written in standard English?

Reviewer #1: Yes

Reviewer #3: Yes

6. Review Comments to the Author

Reviewer #1: (No Response)

Reviewer #3: (No Response)

7. PLOS authors have the option to publish the peer review history of their article (what does this mean?). If published, this will include your full peer review and any attached files.

Reviewer #1: No

Reviewer #3: No

---

## [Editor Report · Acceptance letter]

29 Dec 2023

PONE-D-22-34227R1 

PLOS ONE

Dear Dr. Puthettu, 

I'm pleased to inform you that your manuscript has been deemed suitable for publication in PLOS ONE. Congratulations! Your manuscript is now being handed over to our production team.

Kind regards, 

on behalf of

Dr. Mohamad Khair Abou Chaar 

Academic Editor

PLOS ONE